# Defocused Image Changes Signaling of Ganglion Cells in the Mouse Retina

**DOI:** 10.3390/cells8070640

**Published:** 2019-06-26

**Authors:** Feng Pan

**Affiliations:** School of Optometry, The Hong Kong Polytechnic University, Kowloon, Hong Kong, China; feng.a.pan@polyu.edu.hk; Tel.: +852-2766-6640; Fax: +852-2764-6051

**Keywords:** ganglion cell, myopia, signaling, gap junction, retina

## Abstract

Myopia is a substantial public health problem worldwide. Although it is known that defocused images alter eye growth and refraction, their effects on retinal ganglion cell (RGC) signaling that lead to either emmetropization or refractive errors have remained elusive. This study aimed to determine if defocused images had an effect on signaling of RGCs in the mouse retina. ON and OFF alpha RGCs and ON–OFF RGCs were recorded from adult C57BL/6J wild-type mice. A mono green organic light-emitting display presented images generated by PsychoPy. The defocused images were projected on the retina under a microscope. Dark-adapted mouse RGCs were recorded under different powers of projected defocused images on the retina. Compared with focused images, defocused images showed a significantly decreased probability of spikes. More than half of OFF transient RGCs and ON sustained RGCs showed disparity in responses to the magnitude of plus and minus optical defocus (although remained RGCs we tested exhibited similar response to both types of defocus). ON and OFF units of ON–OFF RGCs also responded differently in the probability of spikes to defocused images and spatial frequency images. After application of a gap junction blocker, the probability of spikes of RGCs decreased with the presence of optical defocused image. At the same time, the RGCs also showed increased background noise. Therefore, defocused images changed the signaling of some ON and OFF alpha RGCs and ON–OFF RGCs in the mouse retina. The process may be the first step in the induction of myopia development. It appears that gap junctions also play a key role in this process.

## 1. Introduction

Myopia is a growing major public health problem, affecting more than 40% of individuals over the age of 12 years in the United States [1] and more than 80% of people in Hong Kong [2,3]. High myopia is a predisposing factor for retinal detachment, myopic retinopathy, and glaucoma, which can lead to loss of vision and blindness [4,5].

Ocular refraction depends primarily on axial length, corneal curvature, lens power and anterior chamber depth [6,7]. Emmetropia occurs when distant images focus on the retinal photoreceptors. During childhood, emmetropization is an active process, in which the expanding eye is adjusted to match the powers of the cornea and lens and tends to result in emmetropia. Any failure of emmetropization results in refractive errors [8]. In myopia (nearsightedness), the eye is relatively long for the optical power of the cornea and lens, and distant images focus in front of the photoreceptors. In hyperopia (farsightedness), distant images focus behind the photoreceptors. The emmetropization process is regulated by defocused image associated with eye growth and refraction in animal research [9,10]. There are also studies examining the human choroid and retinal biophysical activity that had confirmed the findings from animal research that optical defocus both in short-term [11,12,13,14] and long-term [15,16] influenced the mechanisms that would lead to emmetropization /refractive error development.

It is important to determine whether the retina alone could detect the sign of defocus during this visually guided refractive development period [17]. In addition, it remains unclear if the defocused image effects on RGCs’ signaling account for either emmetropia or refractive errors. Recent evidence provided by human electrophysiological studies [14,18] indicated the contribution of the retina in decoding positive and negative defocus. We hypothesis that the first step in progression to ametropia begins within the retina and this study aimed to determine if exposure to defocused images could change the signaling of RGCs in the mouse retina. The process and mechanism of events in the retina will be analyzed from this initial point of myopia development.

There are at least 30 channels or different RGC types in the mouse retina that communicate retinal visual information to the brain [19,20,21]. ON and OFF responses are the most significant visual features encoded by RGCs [22,23]. The separation of ON and OFF pathways is a fundamental principle of retinal organization [24]. RGCs obey the stratification rules, such that OFF RGCs receive input from OFF bipolar cells in sublamina a of the inner plexiform layer (IPL), and ON RGCs receive input from ON bipolar cells in sublamina b, ON/OFF types of RGCs are bistratified [25]. Alpha(α) RGCs are characterized by their large cell bodies, stout dendrites and large mono-stratified dendritic arbors under microscopy. Alpha RGCs play an essential role in visual processing in the retina and transfer the first signals of vision stimulus to the brain [26,27]. Therefore, this study focused on the α RGCs.

Recent research has shown that mice carrying a mutation that blocks the ON channel developed deprivation myopia more rapidly and to higher levels than wild-type mice [28]. Observation of the chicken model indicated that ON activity tends to inhibit myopia, and OFF activity tends to inhibit development of hyperopia, although the mechanism remain unclear [29,30]. Due to the ON/OFF pathway’s effect on development of ametropia, retinal signals are strong candidate for mediating the retina-to-sclera signaling pathway in refractive development and by virtue of retina signal intervals linked to human myopia, may be involved in human myopia. These results support our hypothesis that retinal signal analysis has great promise for defining the signaling mechanisms modulating refractive development.

In this study, differently defocused images projected in front of or behind the mouse photoreceptors were used to test our hypothesis. In addition, as accommodation also has a complex role in myopia development, by directly projecting defocused images on the retina, its effects can be excluded. We find that defocused images change the signaling of RGCs in the mouse retina. The process might be the first step to induce myopia development. Amacrine cells may also play an important function in increasing the signal-to-noise ratio.

## 2. Methods

### 2.1. Ethics Approval

All animal procedures were approved by the Animal Subjects Ethics Sub-Committee (ASESC) of the Hong Kong Polytechnic University and comply with the Guide for the Care and Use of Laboratory Animals published by the National Institutes of Health.

### 2.2. Retina Preparation

Adult mice (postnatal day 42–90) C57BL/6J wild-type (WT), both sex, n = 41; Kcng4-YFP (6–8 weeks, n = 11) mice [31], which labeled α-RGC (Appendix A) were used in the study. The animals were maintained in a 12 h–12 h day–night cycle, and all experiments were performed during daylight hours. The mice were anaesthetized deeply with an intraperitoneal injection of ketamine (Vedno, St. Joseph, MO, USA) and xylazine (Akorn, Decatur, IL, USA) [80 and 10 mg (kg body weight)^−1^, respectively], and lidocaine hydrochloride (20 mg mL^−1^, Sigma-Aldrich, St. Louis, MO, USA) was applied locally to the eyelids and surrounding tissue. Eyes were removed under dim red illumination and hemisected anterior to the ora serrata. Anterior optics and the vitreous humor were removed, and the resultant retina–eyecup with sclera attached, either whole or in sections, was placed in a super-fusion chamber. For patch recordings, retinas were dissected into four equal quadrants and attached to a modified translucent Millicell filter ring (Millipore, Bedford, MA, USA). The flattened retina were superfused with oxygenated mammalian Ringer solution, pH 7.4, at 32 °C [32]. The anaesthetized animals were killed by cervical dislocation immediately after the enucleations.

### 2.3. Full Field Light Stimulation

A green (525 nm) light-emitting diode delivered uniform full-field visual stimulation on the surface of the retina. The intensity of the square-wave light stimuli was calibrated with a portable radiometer/photometer (Ealing Electro-Optics, Holliston, MA, USA) and expressed in terms of the time-averaged rate of photoisomerizations per rod per second (Rh∗/rod/s). Light intensities were calculated assuming an average rod density of 437,000 rods mm^−2^ [33] and quantum efficiency of 0.67 [34].

### 2.4. Single Cell Patterned Light Stimulation

A green organic light-emitting display (OLEDXL, Olightek, China; 800 × 600-pixel resolution, 85 Hz refresh rate) was controlled by an Intel Core Duo computer with a Windows 7 operating system. In our setup, for a Nikon 40x water-immersion objective (CFI Apo 40XW NIR, NA = 0.8), the area of retina that received patterned light stimuli was 250 µm in diameter. Under the 40x objective, the 15 µm diameter pixels of the OLED were 0.25 µm/pixel on the retina. Spatial frequency stimuli generated by PsychoPy (PsychoPy, RRID:SCR_006571) [35] were carefully projected onto the outer segments of photoreceptors layer. The background light intensity was 937 isomerizations Rh*/rod/s. At this level of background illumination, the rod pathway has been shown to be saturated and the cone pathway to mediate the light response [36].

### 2.5. Electrical Recording

Extracellular recordings were obtained from RGCs (41 WT mice) in all retinal quadrants by using whole-cell recordings. Whole-cell recordings were performed by using an Axopatch 700B amplifier connected to Digidata 1550B interface and pCLAMP 10 software (Molecular Devices). Cells were visualized with near infrared light (>775 nm) at ×40 magnification with a Nuvicon tube camera (Dage-MTI, Michigan City, IN, USA) and differential interference optics on a fixed-stage microscope (Eclipse FN1; Nikon, Tokyo, Japan). Retina were superfused at a rate of 1–1.5 mL min^−1^ with a Ringer solution composed of (mM): 120 NaCl, 2.5 KCl, 25 NaHCO_3_, 0.8 NaHPO_4_, 0.1 NaH_2_PO_4_, 1 MgCl_2_, 2 CaCl_2_ and 5 D-glucose. The bath solution was continuously bubbled with 95% O_2_–5%CO_2_ at 32°C.

Electrodes were pulled to 5−7 MΩ resistance, with internal solutions consisting of (mM): 120 potassium gluconate, 12 KCl, 1 MgCl_2_, 5 EGTA, 0.5 CaCl_2_, 0.2GTP and 10 HEPES (pH adjusted to 7.4 with KOH). This internal solution was used in experiments where spiking was not blocked. Spike trains were recorded digitally at a sampling rate of 20 kHz with Axoscope software, with sorting by using Off-line Sorter (Plexon, Dallas, TX, USA) and NeuroExplorer (Nex Technologies, Littleton, MA, USA) software.

### 2.6. Data Analysis

Intensity–response profiles for individual cells were generated by tabulating spike counts or current amplitudes in 1000 ms bins before, during and after the presentation of a stimulus of 1000 ms duration with light intensities varying over 5 log units. The number of light-evoked ON and OFF spikes of RGCs or current amplitudes were calculated by a subtraction of the background spike or current activity from those evoked by the light stimulus onset and offset, respectively. In total, 53 cells were classified as sustained or transient, based on spike frequency parameters as previously described [37].

Averaged response data were then normalized and plotted against the intensity of the light stimuli with Origin 2017 software (OriginLab, Northampton, MA, USA). Data points were fitted by the classic Michaelis–Menten equation [38,39,40], as follows
$$R=\frac{R_{max}I^{a}}{I^{a}+\sigma^{a}}$$
where R is the measured response, Rmax the maximal response, I the stimulus intensity, σ the light intensity that produces a response of 0.5 Rmax, and a is the Hill coefficient. Cells whose intensity–response functions showed a fit of r^2^ < 0.5, probably because of deterioration of recording or adaptation changes during the stimulation series, were excluded from the analysis. Response thresholds for individual cells were taken as 5% of the maximal spike frequency. Population histograms of response thresholds were fitted by non-linear regression using Origin 2017 software. All data are reported as means ± SEM. Statistical significance (*p* < 0.05) was determined using Student’s paired t test.

### 2.7. Immunocytochemistry

After 4% formaldehyde fixation, the tissues were rinsed extensively with 0.1 M phosphate buffer (PB), pH 7.4, and blocked with 3% donkey serum in 0.1 M PB with 0.5% Triton-X 100 and 0.1% NaN3 overnight. The primary antibodies (goat anti-ChAT, 1:100. Millipore Cat# AB144P, RRID:AB_2079751) were diluted in 0.1 M PB with 0.5% Triton-X100 and 0.1% NaN3 containing 1% donkey serum [41]. The tissues will then be incubated for 3–7 days at 4°C and, after extensive washing, incubated in secondary antibodies overnight at 4°C. After washing with 0.1 M PB the tissues were mounted in Vectashield (Vector Laboratories) for observation. The secondary antibodies used were donkey anti-goat Cy-5 (1:200) (Jackson ImmunoResearch Laboratories, West Grove, PA). Neurobiotin was visualized with Cy3-conjugated streptavidin (Jackson ImmunoResearch Laboratories, West Grove, PA). Images were acquired on a Zeiss LSM-800 (Zeiss, Thornwood, NY, USA) confocal microscope using a 40× objective (N.A. 1.3).

## 3. Results

### 3.1. Projecting Defocused Images on Photoreceptors under Microscopy

The design of the experiment was to project defocused images in front or behind the outer segments of photoreceptors to mimic plus and minus defocus similar to myopia or hyperopia (Appendix A). At the same time, the magnitude of refractive errors generated by the defocused image projected through microscopy also needed to be calculated precisely.

According to our experiment, the refractive errors of mice changed from 5.38 to 7.18 diopters from week 6 to week 8. During that time period, the axial length elongated by around 10 µm. The results are similar to those reported previously [42]. As it has been calculated that axial elongation of 5 µm could induce 1 Diopter (D) refractive error [6], 100 µm defocus would be expected to induce plus or minus 20 diopters refractive error under microscopy depending on the direction of defocus. To confirm this calculation, the +20 D Lens was placed in front of an organic light-emitting diode (OLED) to defocus the image, and the focus of microscope was adjusted to be 100 µm in front of the retina to refocus the shifted image under 40X water lens. The system could then artificially generate any different plus and minus magnitude of defocus in the whole retina. The custom-made light-projected system was made to project a programmed light spot onto a single cell receptive field and precisely focus on the photoreceptor cells layer.

Projecting defocused images under microscope has the following advantages. Firstly, the different magnitude of refractive errors can be accurately controlled so that plus and minus defocused images can be easily achieved. Secondly, with the help of filters in the microscope, various light intensities can also be controlled. Patterned images can also be produced by available software.

### 3.2. The Light Intensities of the Defocused Image Activate the Cone Pathway

Alpha RGCs, which exist in many species, including humans, play an essential role in visual processing. RGCs show a divergence of their rod pathways in dark-adapted conditions [43], with some RGCs showing pure cone-mediated responses. 

According to our previous experiments, these alpha RGCs are low-intermediated sensitivity RGCs [19,44]. Cell light responses are saturated after light stimuli above 1000 Rh* per rod per second. Therefore, light threshold sensitives of RGCs were tested first. ON and OFF alpha RGCs were easily recognizable under the 40 water immersion Lens- by their large, specific morphologic cell bodies. In addition, ON and OFF α-RGCs could be further visualized by Neurobiotin injection after recording (Figure 1C,F). Double labeling with anti-ChAT antibody showed their large mono-stratified dendritic arbors. ON and OFF alpha RGCs showed increasing spikes to a full-field 525 nm light stimulus (Figure 1A,D). The light intensity–response function of the ON and OFF α-RGC had a threshold sensitivity of 17.4 and 7.02 Rh* /rod/s (Figure 1B,E).

During the following defocus experiments, there was variation of light intensities. For example, light intensities increased from 1.53 × 10^5^ Rh*/rod/sec to 1.69 × 10^5^ Rh*/rod/sec with 125 µm 0 cycles/degree image defocused from −20 to +20 diopters. Similarly, light intensities increased from 4.99 × 10^4^ Rh*/rod/sec to 5.25 × 10^4^ Rh*/rod/sec with 125 µm 0.002 cycles/degree image defocused; 0.0067 cycles/degree image defocused will change light intensity from 3.74 × 10^4^ Rh*/rod/sec to 3.85 × 10^4^ Rh*/rod/sec; 0.05 cycles/degree changed from 3.2 × 10^4^ Rh*/rod/sec to 3.46 × 10^4^ Rh*/rod/sec (Figure 2A). In addition, increasing spatial frequencies will decrease light intensities. However, the light intensities of different spatial frequencies applied in the experiment varied from10^4^ to 10^5^ Rh*/rod/s (1–2 log units above background). At these light intensities, the RGCs’ light responses are expected to be activated in the cone pathway where is responsible for defocused images.

Projecting defocused image on retina will change the area and edge of image imposing on the RGC layer. This will bring the concern about change of RGC receptive field. In the experiment, imposing defocused image will generate linear increasing image size with blurred edge. For example, the 125 µm light spot defocused ±5 diopters will increase this 125 µm light spot to 150 µm; 175 µm in ±10 diopters; 225 µm in ±20 diopters (Figure 2B). With increasing defocused power, the edge of images also became blurred and hard to define, especially in ±20 diopters defocused image. 

To minimize the effect of changed receptive field under the defocused images, we adjusted the image area projected on the RGC layer by programming to rend them identical (for example, 125 µm in diameter) with respect to different defocus powers. Thus, concern with variation in the receptive field accompanied with the defocused image will be minimized.

### 3.3. RGCs Showed Decreasing Spikes to Smaller Light Spot Stimuli

Under the 40 water immersion Lens, the area of retina that could receive light stimuli was up to 250 µm in diameter (150 × 200 µm area). Therefore, a 125 µm light spot was projected onto the outer segments of photoreceptors. The full spot light intensities were 1.69 × 10^5^ Rh*/rod/s. The background light intensity was 937 Rh*/rod/s. 

The ON α-RGC had a response to a 125 µm light spot. The peristimulus time histogram of the ON α-RGCs responded to the presentation of a 1 s full-field light stimulus with a probability of spikes around 1.4 (Figure 3A). The RGC showed a decreased probability of spikes (from 1.4 to 0.42) with a 75 µm light spot (Figure 3B) and almost lost the light response with a 50 µm light spot (Figure 3C). However, if a 125 µm full light spot was defocused under the microscope equal to −20 diopters, the probability of spikes decreased from 1.4 to 1.1 (data not shown). Under the 40 water immersion Lens, 125 µm in diameter light spot can induce a stable light response under the system. Therefore, a 125 µm light spot was used in the subsequent experiments.

### 3.4. Different Spatial Frequencies Image Could Change the Signaling of α-RGCs

As in myopia patients, defocused images generate blurred images on the retina. However, the blurred and defocused images are two different concepts. Blurred images can be precisely projected on retina whereas defocused images are not. By projecting different spatial frequency images, RGCs were observed to see whether their light responses were similar between the defocused images and different spatial frequencies images. Altering the spatial frequency of the projected image would only simulate image blur without providing any cue to the sign of image blur induced by defocused images (given that both myopic and hyperopic defocused images would lose high spatial frequency content regardless of the sign of defocus).

First, a 125 µm 0 cycles/degree light spot with 1 s light stimulus (I = 1.665 × 10^5^ R*/rod/sec) was projected on an ON α-RGC (Figure 4A). Then, about −20D defocused image in the peristimulus time histogram of the ON α-RGC showed that the probability of spikes decreased from 0.65 to 0.25; 95% confidence limit were similar (Figure 4B). The ON α-RGC response to 125 µm 0.002 cycles/degree light showed that the probability of spikes decreased from 0.65 to 0.62 while the 95% confidence limit dramatically increased from 0.25 to 0.5 (Figure 4C). However, ON α-RGCs lost their light response to a 125 µm, 0.0067 cycles/degree light stimulation (Figure 4D). The experiment showed that RGCs responded differently between the defocused images and different spatial frequencies images (results from Figure 4B–D).

### 3.5. Defocused Images Altered the Signaling Responses in RGCs

By projecting an image in front of or behind the outer segments of the photoreceptor layer, a varied magnitude of minus and plus refractive error could be artificially produced.

The ON α-RGC responded differently to plus and minus defocused 125 µm 0.002 cycles/degree light spots (1 s light stimulus I = 5.09 × 10^4^ Rh*/rod/sec). The peristimulus time histogram of the ON α-RGC showed that the probability of spikes dropped from 2.05 to 1.25 with a −10D defocused image and the probability of spikes dropped from 2.05 to 0.94 with a +5D defocused image (Figure 5A–C).

The OFF α-RGC responded to 125 µm 0.002 cycles/degree light spot (1 s light stimulus I = 5.09 × 10^4^R*/rod/sec). The peristimulus time histogram of the OFF α-RGC showed that the probability of spikes decreased from 1.4 to 1.2 under −10D defocus and the probability of spikes decreased to 0.6 with +5D defocus.95% confidence limits were similar in these trials (Figure 5E–G).

Similarly, the ON–OFF RGC showed that the probability of spikes of ON response decreased from 0.44 to 0.27 with −10D defocus, and to 0.22 with +5D defocus. Interestingly, the OFF response of the ON–OFF RGC dropped from 0.47 to 0.44 under −10D and to 0.37 under +5D defocus. The ON responses of the RGC were more sensitive than OFF response to defocus (Figure 5 I–K). 

The probability of spikes of these ON, OFF and ON–OFF RGCs can be recovered after image refocused (Figure 5D,H,L).

### 3.6. Some RGCs Responded Differently to Plus and Minus Defocus

It is interesting to observe that RGCs could distinguish between plus and minus defocused images and respond differently.

The result showed that OFF transient RGCs (13 of 25 OFF RGCs, around 52%) showed different responses to the magnitude of plus and minus defocused images. The probability of spikes of OFF transient RGC responded to defocused images up to −16 or −20 diopters. However, the cell only responded up to a +5 diopters defocused image (Figure 6A).

Seven of 25 (28%) OFF transient RGCs showed similar responses to the magnitude of plus and minus defocused images (Figure 6B), but five of 25 (20%) OFF transient RGCs showed more response to plus defocused image (Figure 6C).

With respect to OFF sustained RGC, of the 13 cells, 7 (54%) showed a higher probability of spikes of cell responses to plus defocused images of +16 or +20 diopters versus no responses to less than −10 diopters defocused images. The remaining OFF sustained RGCs showed no difference between the magnitude of plus and minus images.

ON transient RGCs seem to be insensitive to defocused images, with seven (100%) showing almost no difference in response to plus and minus up to 20 diopters defocused images.

ON sustained RGCs also seemed to be insensitive to defocused images. Three of eight (37.5%) showed no difference in response to plus and minus up to 20 diopters defocused images while the remaining five cells (62.5%) showed more response to minus defocused images.

Five of the ON–OFF RGCs showed a relatively similar ON and OFF responses to plus and minus defocused images (Figure 6D). 

### 3.7. ON and OFF Alpha RGCs’ Response to Defocused Image Was Affected by Gap Junctions. Amacrine Cells May Contribute to These Effects Through Gap Junctions

The effect of gap junctions on RGCs response to defocused image was explored. The ON α-RGC response to 0.002 cycles/degree (1 s light stimulus, I = 5.09 × 10^4^ R*/rod/sec) was recorded with focused and −10D defocused images. The peristimulus time histogram of the ON α-RGC showed that the probability of spikes decreased from 0.87 to 0.79 (Figure 7A,B).

However, after application of the gap junction antagonist, 25 µM 18-beta-glycyrrhetinic acid (18-β) [45], the probability of spikes decreased from 0.87 to 0.52 with focused images (Figure 7C). In contrast, the probability of spikes decreased from 0.79 to 0 with a −10D defocused image (Figure 7D).

As ON alpha RGCs are only coupled to amacrine cells in the mouse retina [44], it appears that amacrine cells make a contribution to signaling of RGCs to the defocused images.

The OFF α-RGC responded to both focused and -10D defocused images. The peristimulus time histogram of the OFF α-RGC showed that the probability of spikes decreased from 1.58 to 1.03 (Figure 7E,F). After 25 µM 18-β application, the probability of spikes decreased from 1.58 to 0.85 (Figure 7G).

The probability of spikes reduced from 1.03 to 0.51 with a -10D defocused image. However, the 95% confidence limits increased to around 0.5 (Figure 7H), indicating that the cell barely responded to the defocused image in the presence of gap junction blocker.

As OFF alpha RGCs are coupled to other OFF alpha RGCs and amacrine cells, it once again suggests that amacrine cells play a role in the response of RGCs to defocused images.

### 3.8. ON Alpha RGCs Showed Decreased Responses to Defocused Images After the Application of Gap Junction Blocker

ON alpha transient RGCs showed a significantly decreased response to both plus and minus defocused images two minutes after gap junction antagonist, 25 µM 18-β application. The average of the probability of spikes decreased by around 50% (Figure 8A).

ON alpha sustained RGCs showed almost no difference in response to plus and minus defocused images at +16D and −20D, but 5 min after 25 µM 18-β application, the cells showed no response to defocused images at +5D and −15D (Figure 8B). Apparently, gap junctions affected the cell response to the defocused image.

### 3.9. ON–OFF RGCs Showed Different Responses to Defocused Images and Spatial Frequencies. Amacrine Cells Might Contribute to the Effects Through Gap Junctions

Gap junction antagonist, 25 µM 18-β was used to block gap junctions between ganglion cells and amacrine cells [45]. Under 125 µm, 0 cycles/degree light spot (1 s light stimulus, I = 1.665 × 10^5^ R*/rod/sec) and defocused −20D image, the peristimulus time histogram of the ON response of the cell showed that the probability of spikes decreased from 1.1 to 0.35, while the OFF response dropped from 0.63 to 0.28 (Figure 9A,B). Under 125 µm, 0.05 cycles/degree light stimuli, the probability of OFF spikes decreased to 0.17, while at the same time, the ON response was lost (Figure 9C). Under a 125 µm 0.0067 cycles/degree light stimuli, the probability of spikes of OFF decreased to 0.3 and the ON response was still not present (Figure 9D). However, with a 125 µm, 0.002 cycles/degree light stimuli, the probability of OFF spikes increased to 1.1, which was similar to 0 cycles/degree light stimuli and ON responses of the cell increased to 0.57(Figure 9E).

In next step, gap junction antagonist, 25 µM 18-β was applied, and the probability of ON spikes of the cell to a 125 µm, 0.002 cycles/degree light stimuli decreased from 1.1 to 0.6 with the 95% confidence limit increasing from 0.35 to 0.4. However, the OFF response disappeared (Figure 9F).

Gap junction antagonist also changed the cell light response to 125 µm, 0 cycles/degree light stimuli. The probability of ON spikes (Figure 9G) was similar to the control (Figure 9A), both were 1.1. However, the 95% confidence limit increased from 0.2 to 0.35 (Figure 9G). These results showed that the function of amacrine cells might increase the signaling and decrease the background noise.

## 4. Discussion

Despite the major public health impact, the etiologies of refractive errors and myopia are poorly understood. Current research indicates that optical defocus and image blur alter eye growth and refraction [9]. The retina governs refractive development. However, a comprehensive understanding of the signaling of RGCs that accounts either for normal refractive development or for refractive errors has remains elusive. Understanding the signaling of RGCs may help the development of clinically acceptable therapies to prevent myopia onset or to slow its progression.

### 4.1. Mouse Eye Growth Models Can Provide a Powerful Means to Study Refractive Development

Mouse experimental myopia models provide a powerful means to study refractive development with proven utility for mammalian eye growth and human refractive development [6,42].

The experimental myopia mouse model has several advantages: (1): The mouse is currently the most extensively studied mammalian model for human diseases [46] and there is considerable knowledge about the retinal circuit [47]; (2): The myopia mouse model has a similar sclera structure and fibroblasts to humans [48,49]; (3): Numerous viable knockout mouse models have been developed.

Retinal signaling is a strong candidate for mediating the retina to sclera signaling pathway in refractive development [50,51], ultimately leading to myopia. However, there is evidence that retinal ganglion cells and even the retina is not needed for eye growth [17,52,53,54]. The results of this study imply that the retinal signaling mechanism might involve in modulating refractive development.

To precisely present a defocused image at the outer segments of photoreceptors in the mouse retina, it is essential to calculate the defocused distance and its equivalent refractive error. In our calculation and setup, 5 µm defocus made by microscope could induce refractive error equal to plus or minus 1 diopter, depending on the direction of Lens movement [6]. In addition, the link between RGCs and cell types related to myopia development still remains unclear. In this study, amacrine cells seem to be a promising candidate for transfer of signaling from RGCs to other cells and tissues, possibly including the sclera which are involved in increasing axial length.

### 4.2. Defocused Image Changes the Signaling of Ganglion Cells in the Mouse Retina

Most research related to the pathogenesis of refractive errors has addressed myopia and hyperopia. Epidemiologic studies typically survey conventional factors which may be related to myopia, such as family history, education, intelligence, and socioeconomic status, etc. [55,56], but study outcomes are not conclusive and often contradictory [57,58,59]. Although several interventions, including orthokeratology [60], atropine therapy [61] and mixed defocused lenses [62] can slow down axial growth significantly, clinically effective therapies to fully normalize eye growth are not available. Understanding the etiology of myopia remains a major goal of the National Eye Institute strategic plan and Asian countries.

The study’s results indicate that defocused image may be the first step in induction of myopia development. The evidence from animals implicated the role of the visual image on the retina in refractive development as the main controlling mechanism [63]. Optical defocus produced by the wearing of defocusing spectacle lenses to shift the image plane in front of or behind the retina has been shown to compensate for changes in eye growth to reposition the retina at the image plane in chicks [64], mouse [65], tree shrews [66], marmosets [67] and monkeys [68,69]. These animal optical defocus results resemble those of human clinical studies [11,12,70,71,72]. Recent evidence from the human eye suggests a regional choroidal response [73] and retinal response [18] to defocus. Much evidence localized the visual mechanism regulating eye growth largely to the retina itself [63,74]. The current study demonstrated that defocused image decreased probability of spikes in ON, OFF and ON–OFF RGCs. Both plus and minus defocused images decreased the probability of spikes which indicated alteration of the signaling of RGCs.

The study has attempted to explain the direct mechanism that is most relevant to myopia or hyperopia development. Importantly, retinal neurotransmitters such as dopamine [75,76], VIP [77], and retinoic acid [78,79], have all been independently implicated as potentially mediating eye growth by having an effect on RGCs.

In this study, it was interesting to observe that some OFF transient RGCs and ON sustained RGCs showed different responses to plus and minus defocused images. Given that animal studies [28,80,81,82,83] suggest that selective effects of ON and OFF pathway activation on eye growth and refractive error development and evidence from the human eye regarding the contribution of ON/OFF retinal pathways to the mechanisms of emmetropisation [84]. The disparity of response of signaling of RGCs may be the first step for RGCs to distinguish between the plus and minus defocused images. Recently, Dr. Schwartz’s group found an “ON-delayed” RGC with high sensitivity to high spatial frequency patterns presented over large receptive field, which might be the defocus detector in mouse retina [85,86]. However, a study showed that RGCs did not respond differently to defocus directions by using multi-electrode array in chicken retinas [87]. More types of RGCs are needed to be tested in the further study.

### 4.3. Different Spatial Frequencies Can Also Change the Signaling of Ganglion Cells

The mammalian visual system is able to simultaneously create neural representations of any visual scene on several different scales. In the mouse retina, RGCs with different sized receptive fields could capture the presence of borders or edges at different scales in the region of the retina over which visual signals influence the activity of that neuron [88]. Sixty years ago, a theory of form perception in which the visual system would use neurons with different receptive field sizes to create a series of neural representations on different scales was proposed [89]. Their theory, which has been designed to numerous spatial frequency which varied in contrast, orientation, and spatial phase has been widely used to test form reception and visual acuity.

At the cell level, it was wrong to hypothesize that the different spatial frequencies could mimic the myopic image projected on the retina. The blurred images can still be precisely projected on retina whereas defocused images are not. Defocused images can also change the projected area with slightly different light intensities (Figure 2). However, the signaling variation of RGC responses to different spatial frequencies can allow exploration of the mechanism of electrophysiological signaling at the single cell level. Research suggests human retinal response variations with optical defocus and spatial frequency content of the stimulus [18]. Recently, human local regional choroidal was shown to have response to myopic defocus [13]. These evidence might imply the possible myopia control interventions.

In this study, ON and OFF RGCs normally showed an increased probability of spikes to low spatial frequencies (few cycles within each degree of visual angle). That is, low spatial frequency light spots were likely to act at, pass across and/or stimulate more amacrine cells to synthesize or release mediator feedback to RGCs through gap junctions. Recent work (2018 ARVO 1856-C0160) has shown that the cell needs a coupling network to fire and code visual information. Therefore, defocused images alter the coupled amacrine cells to the RGCs network. This altered coupling may provide key inhibition to the signaling of RGCs to both focused and defocused images.

### 4.4. Amacrine Cells Play an Important Function in Increasing Signaling and Decreasing Background Noise

The overall hypothesis for this research is that defocused images alter RGC signaling, which induces subsequent transcriptome level changes in the retina and that these electrical signatures can act as important retinal mediators of refractive development. The findings which define RGCs’ signaling alteration induced by defocused image, support this hypothesis. After the application of gap junction blocker, the ON alpha RGCs which only coupled to amacrine cells, their responses to defocused images changed. This indicates that amacrine cells play a key role in the signaling of RGC responses to defocused images. In addition, some OFF transient RGCs responded differently to plus and minus RGCs. This raises an interesting question of how RGCs sense the different defocused images. It could be because of OFF RGCs coupled to other OFF RGCs and amacrine cells. Plus, and minus defocus images apparently projected different size and light-intensities’ images on the retina, because the different coupled amacrine cells and RGCs are activated-thereby allowing RGCs to sense the difference between plus and minus defocused images.

In ON–OFF RGCs, defocused images and spatial frequency images had different effects on the ON and OFF light responses (Figure 9). The ON response decreased dramatically after −20D defocused image exposure in the ON–OFF RGC. However, with decreased spatial frequency, the ON response still presented. The reason might be due to more coupled ACs being activated with decreased spatial frequency. The gap junction blocker decreased both the ON and OFF responses and also increased the background noise. The disparity of the ON and OFF response in ON–OFF RGCs might act through ACs activated by the plus and minus defocused images. The reason might be that plus and minus defocused images had small but difference in light intensities (Figure 2) of images projected onto the retina.

We believe that the study of signaling of RGCs has the potential to achieve much needed mechanistic insights into refractive development. Ultimately, the manipulation of signaling of RGCs could be an effective approach to control myopia in children.

## Figures and Tables

**Figure 1 cells-08-00640-f001:**
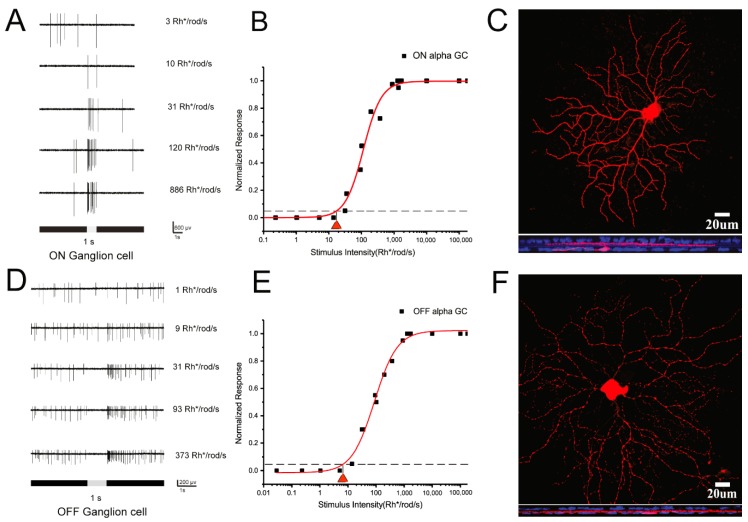
Dark-adapted ON and OFF α-RGCs show low-intermediate light threshold sensitivities. **A** and **D**; ON and OFF RGCs show increasing spikes to full-field 525 nm light stimuli of increasing intensities. **B** and **E**; Intensity–response function of the ON and OFFα-RGC, showing a threshold sensitivity of around 10 Rh* /rod/s. **C** and **F**; ON and OFFα-RGCs can be visualized by Neurobiotin injection after recording. Double labeled with anti-ChAT antibody (blue). Scale bar = 20 µm.

**Figure 2 cells-08-00640-f002:**
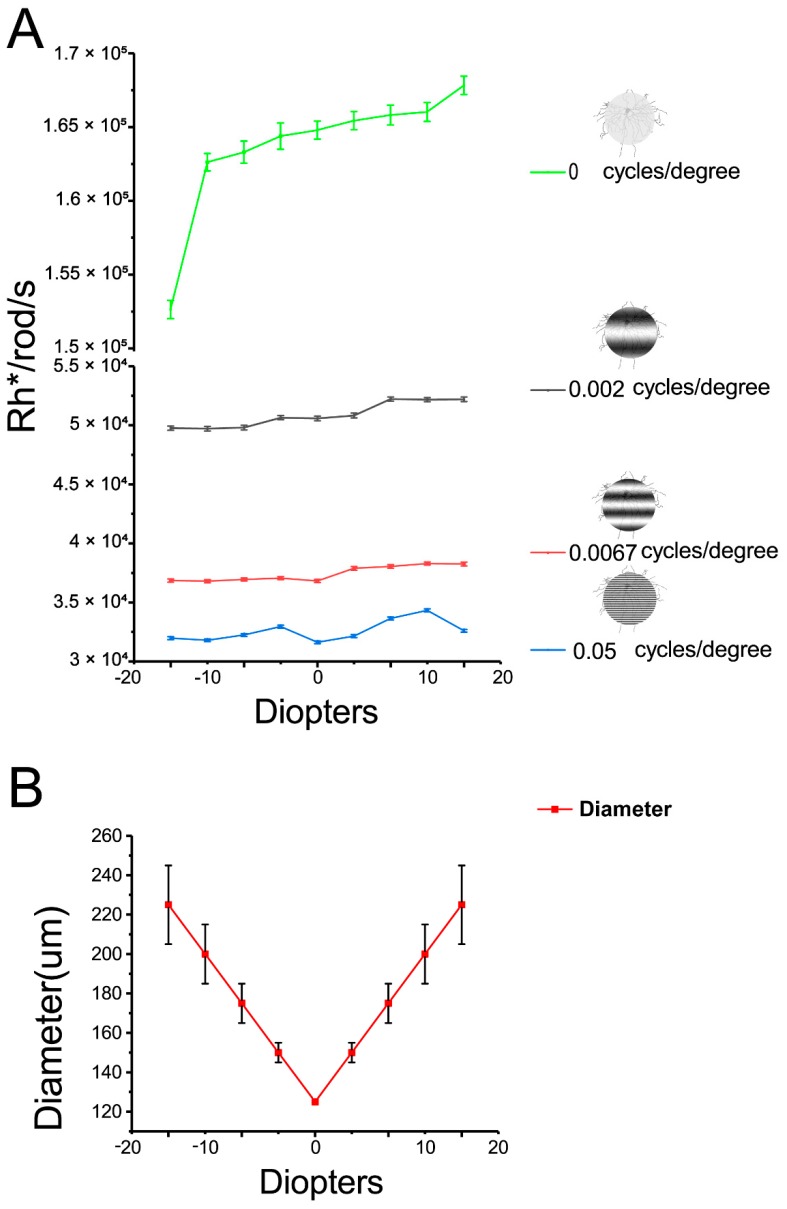
Variance of light intensities and light spot area with defocused image. **A**: Defocused images of a 125 µm 0 cycles/degree from -20 to +20 diopters increased the light intensities from 1.53 × 10^5^ Rh*/rod/sec to 1.69 × 10^5^ Rh*/rod/sec. Increasing spatial frequencies will decrease light intensities. Defocused different spatial frequencies images from −20 to +20 diopters increased light intensities slightly. Defocused images of a 125 µm 0.002 cycles/degree from −20 to +20 diopters changed the light intensities from 4.99 × 10^4^ Rh*/rod/sec to 5.25 × 10^4^ Rh*/rod/sec; 0.0067 cycles/degree from 3.74 × 10^4^ Rh*/rod/sec to 3.85 × 10^4^ Rh*/rod/sec; 0.05 cycles/degree from 3.2 × 10^4^ Rh*/rod/sec to 3.46 × 10^4^ Rh*/rod/sec. **B**: Increasing the diopters of defocused images of a 125 µm light spot results in a linear increase in the diameter of image projected on the RGC layer. With increasing defocused distance, the edge of the image became hard to define, especially with the +20 and −20 diopters defocused images imposed.

**Figure 3 cells-08-00640-f003:**
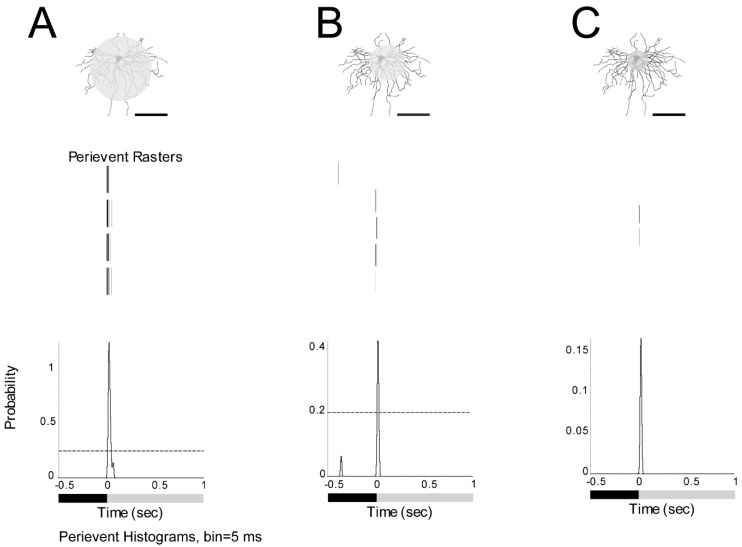
Dark-adapted ON α-RGCs show decreasing spikes to smaller light spot stimuli. **A**: ON α-RGC response to a 125 µm light spot. **B**: The peristimulus time histogram shows the decreased probability of spikes to a 75 µm light spot. **C**: The peristimulus time histogram shows that the cell almost lost the light response to a 50 µm light spot. Therefore, a 125 µm light spot was used in the subsequent experiments. (1 s light stimulation). Scale bar: 75 µm.

**Figure 4 cells-08-00640-f004:**
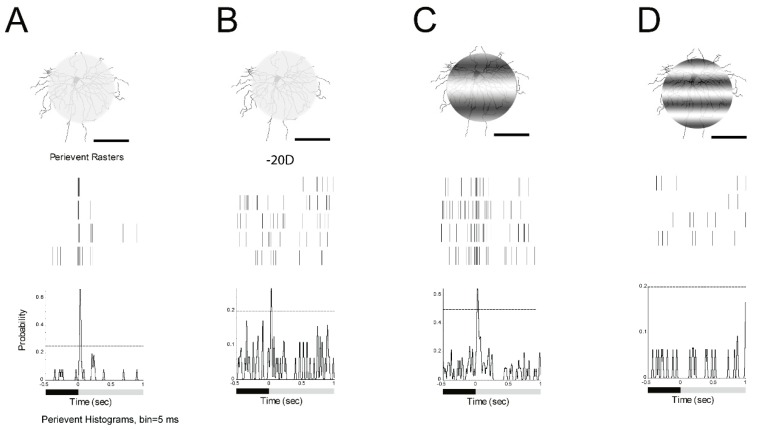
Defocused images and different spatial frequencies can both change the signaling of α-RGCs. **A**: ON α-RGC response to 1 s 125 µm, 0 cycles/degree light spot. Light stimulus (I = 1.665 × 10^5^ Rh*/rod/sec). **B**: ON α-RGC response to 1 s 125 µm, 0 cycles/degree light spot while defocus under microscopy to -20D. The peristimulus time histogram of the ON α-RGC showed that the probability of spikes decreased from 0.65 to 0.25; 95% confidence limits for the conditions are similar. **C**: Peristimulus time histogram of the ON α-RGC response to 1 s 125 µm, 0.002 cycles/degree light showed that the probability of spikes decreased from 0.65 to 0.62. However, 95% confidence limits are similarly increased from 0.25 to 0.5. **D**: ON α-RGCs lost their light response to 1 s 125 µm, 0.0067 cycles/degree light stimulation. Scale bar: 75 µm.

**Figure 5 cells-08-00640-f005:**
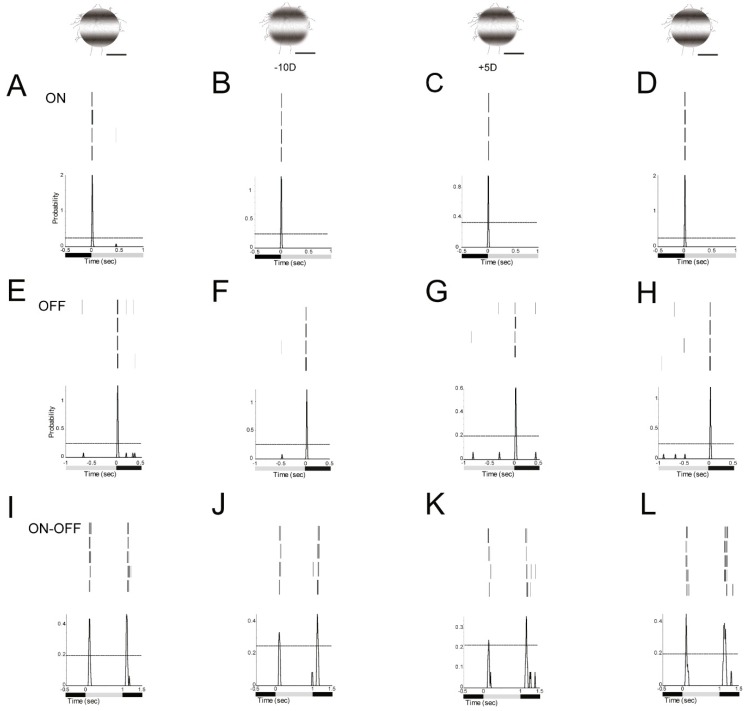
Defocused images altered the signaling responses in RGCs. **A**, **E** and **I**; ON, OFF α-RGC and ON–OFF RGC response to 1 s 0.002 cycles/degree light stimuli (I = 5.09 x10^4^ Rh*/rod/sec). **B**, **F** and **J**; ON, OFF α-RGC and ON–OFF RGC light responses decreased after a defocused image was projected. The peristimulus time histogram of the ON α-RGC (**A**) showed that the probability of spikes decreased from 2.05 to 1.25 (-10D defocus) (**B**) and to 0.94(+5D defocus) **(C)**. Then the probability of spikes recovered with refocused image (**D**). Similar, OFF α-RGC (**E**) decreased from 1.4 to 1.2 (-10D defocus) **(F)** and to 0.6(+5D defocus) (**G**). The probability of spikes recovered with refocused image (**H**). ON–OFF RGC **(I)** ON response decreased from 0.44 to 0.27 (-10D defocus) (**J**) and to 0.22(+5D defocus) (**K**); OFF response decreased from 0.47 to 0.44 (-10D defocus) (**J**) and to 0.37 (+5D defocus) (**K**). Then the probability of spikes recovered with refocused image (**L**). Scale bar: 75 µm.

**Figure 6 cells-08-00640-f006:**
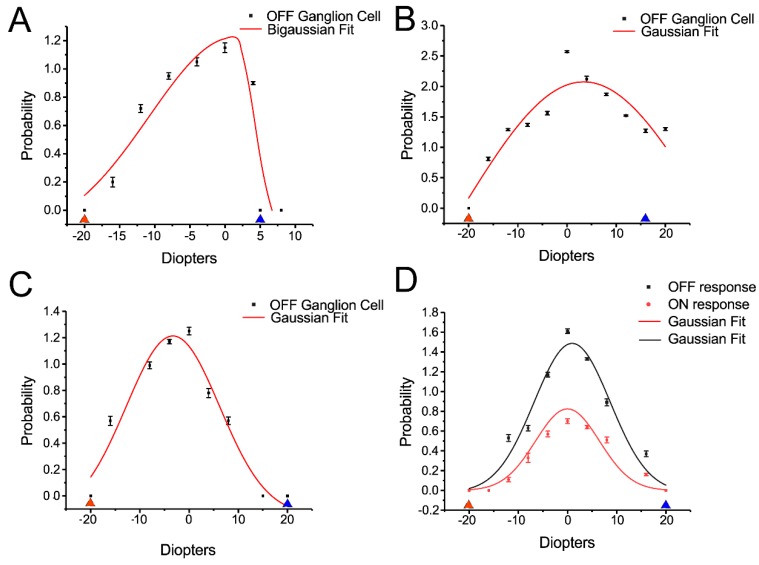
Some RGCs can respond differently between plus and minus defocus. **A:** Some OFF transient α- RGCs showed a different response to the magnitude of plus and minus defocused images. The cells showed no response to defocused image more than +5D (blue triangle) and -20D (red triangle). **B**: Some OFF transient α- RGCs showed similar responses to plus and minus defocused images. The cells showed no response to defocused images more than +16D (blue triangle) and less than −20D (red triangle). **C**: Some OFF transient α- RGCs showed more response to plus than minus defocused images. The cells showed no response to defocused images around −20D (red triangle), however, they still had light response to +20D (blue triangle). **D**: ON–OFF RGCs showed similar of ON and OFF response to plus and minus defocused images. The cell lost light responses at plus and minus 20D.

**Figure 7 cells-08-00640-f007:**
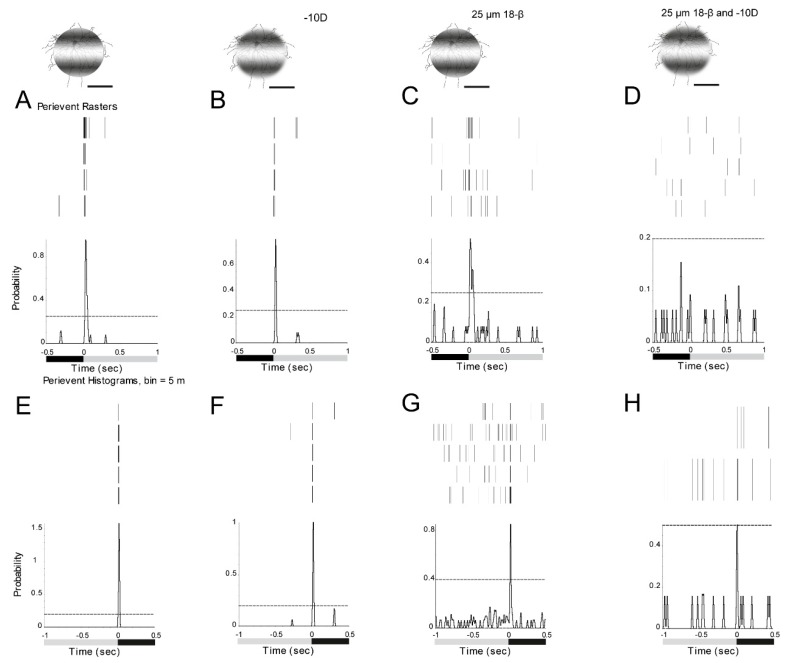
ON and OFF RGCs responses to defocused images can be affected by gap junctions. The ON RGC response can be solely affected by amacrine cells. **A**: ON α-RGC responses to 1 s 0.002 cycles/degree light stimuli (I = 5.09 × 10^4^ Rh*/rod/sec). **B**: The ON α-RGC response to −10D defocused image. The peristimulus time histogram of the ON α-RGC showed that the probability of spikes decreased from 0.87 to 0.79. After gap junction antagonist, 25 µM 18-beta-glycyrrhetinic acid (18-β) application, the probability of spikes decreased from 0.87 to 0.52 (**C**). The probability of spikes decreased from 0.79 to 0 with a −10D defocused image (**D**). **E**: OFF α-RGC response to 1 s 0.002 cycles/degree light stimuli (I = 5.09 × 10^4^ Rh*/rod/sec). **F**: The OFF α-RGC response to a -10D defocused image. The peristimulus time histogram of the OFF α-RGC showed that the probability of spikes decreased from 1.58 to 1.01. After gap junction antagonist, 25 µM 18-beta-glycyrrhetinic acid (18-β) application. The probability of spikes decreased from 1.58 to 0.85 (**G**). The probability of spikes decreased from 1.03 to 0.51 with -10D defocused image (**H**). However, the 95% confidence limit increased to around 0.5. Scale bar: 75 µm.

**Figure 8 cells-08-00640-f008:**
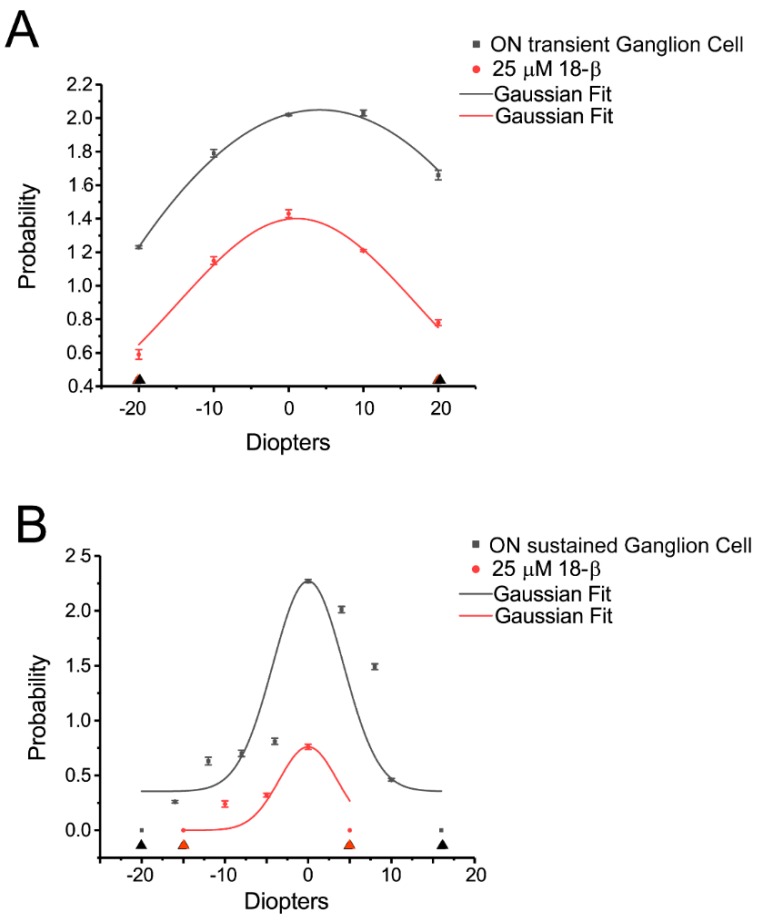
ON alpha RGCs showed decreased responses to defocused image after gap junction blocker application. **A**: ON alpha transient RGCs showed a significantly decreased response (the probability of spikes) to plus and minus defocused images two minutes after gap junction antagonist, 25 µM 18-β application. The cell showed no response to plus and minus defocused images above 20D (black and red triangle). **B**: ON alpha sustained RGCs showed no response to plus and minus defocused images at +16D and -20D (black triangle). Five minutes after 25 µM 18-β application, the cells showed no response to defocused image at +5D and −15D (red triangle).

**Figure 9 cells-08-00640-f009:**
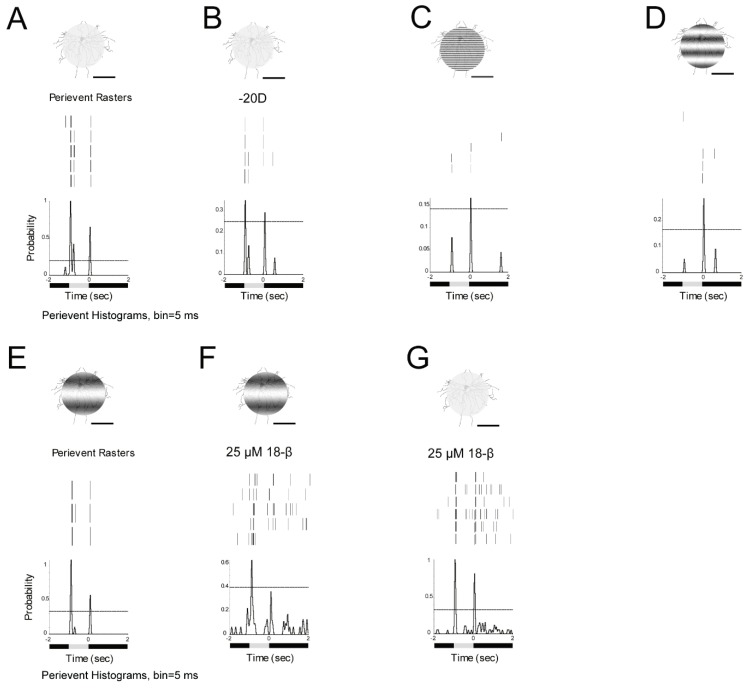
ON–OFF RGCs showed different responses to defocused images and spatial frequencies. Amacrine cells might contribute to the effects through gap junctions. **A**: ON–OFF RGC response to 1 s 125 µm, 0 cycles/degree light spot. Light stimulus (I = 1.665 × 10^5^ Rh*/rod/sec). **B**: ON–OFF RGC response to 1 s 125 µm, 0 cycles/degree light spot with defocus to −20D. The peristimulus time histogram of the ON response showed that the probability of spikes decreased from 1.1 to 0.35. OFF response decreased from 0.63 to 0.28. However, the 95% confidence limit increased from 0.2 to 0.25. **C**: Response to 1 s 125 µm, 0.05 cycles/degree light stimuli showed that the probability of spikes of OFF response decreased to 0.17. ON response was lost. **D**: Response to 1 s 125 µm, 0.0067 cycles/degree light stimuli showed that the probability of spikes of OFF decreased to 0.3. ON response remained absent. **E**: Response to 1 s 125 µm, 0.002 cycles/degree light stimuli showed that the probability of spikes was 1.1 which is similar to 0 cycles/degree light stimuli. However, 95% confidence limit increased to 0.35 from 0.2. **F**: Gap junction antagonist, 25 µM 18-beta-glycyrrhetinic acid (18-β) application changed the cell light response to 125 µm, 0.002 cycles/degree light stimuli. The probability of spikes of ON decreased from 1.1 to 0.6; 95% confidence limit increased to 0.4 from 0.35. **G**: Gap junction antagonist application also changed the cell light response to 1 s 125 µm, 0 cycles/degree light stimuli. The probability of spikes of ON was similar, both are 1.1. However, 95% confidence limits increased from 0.2 to 0.35. These results showed the function of amacrine cells might contribute to increase the signaling and to decrease background noise. Scale bar: 75 µm.

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
