# Peer review of "Defocused Image Changes Signaling of Ganglion Cells in the Mouse Retina"

_cells, 2019, doi:10.3390/cells8070640_

Round 1

Reviewer 1 Report

General

It is an important question how defocus is encoded by retinal neurons and several labs have tried to learn more about the underlying image processing. It was previously found that glucagon amacrine cells in chicks retina express more ZENK protein and ZENK mRNA when the focal plane is in front of the retina, and less when it is behind - but the mechanism seemed to work only in alert chickens. Nothing could be found under anesthesia. It was also attempted to detect differences in retinal ganglion cells responses in vitro, using MEA recordings, when projected movies were defocused in different directions, both in chick and mouse retina

https://www.google.de/url?sa=i&rct=j&q=&esrc=s&source=images&cd=&cad=rja&uact=8&ved=2ahUKEwiiyKmQm6_iAhWCyKQKHZGQAN0Qjhx6BAgBEAI&url=https%3A%2F%2Fpublikationen.uni-tuebingen.de%2Fxmlui%2Fbitstream%2Fhandle%2F10900%2F79499%2FKlaus%2520Graef%2520-%2520Dissertation%2520zur%2520Publikation.pdf%3Fsequence%3D2%26isAllowed%3Dy&psig=AOvVaw00L-c_p7w-n0rYI7Cfh2II&ust=1558617053418387

(I apologize for the long link; furthermore, it is a pity that this PhD work by Dr. Klaus Graef was only available in German)

Another relevant publication is PubMed PMID:19958566 where MEA recordings were done in chicken retinas in vitro but no differences were found in retinal ganglion cells between positive and negative defocus.

There are a few complications:

(1) if defocus is imposed, image magnification and brightness vary, if no Badal system is involved. Both variables may critically affect the responses of the recorded cells without that defocus is the relevant variable

(2) there is suggestive evidence that defocus detection by the retina requires moving images (PubMed PMID: 16289273)

(3) finally, there is evidence that retinal ganglion cells are not needed for retinal control of eye growth. Eye growth compensates for defocus imposed by lenses also when the optic nerve was cut, and myopia can be induced in tree shrews after action potentials were blocked in the retina by TTX. The markers that are changed when defocus is induced are mostly in amacrine cells, suggesting the ganglion cells are transmitting visual information to the brain but do not encode the signals that cause the release of messengers that control choroidal changes and scleral growth.

The author of the current study has performed single whole cell recordings from identified retinal ganglion cells of the mouse in vitro under the microscope. Defocus was imposed by moving the objective of the microscope. A raw calibration was done by adding a 20D lens and check how much the objective had to be moved to refocus the image, if I understand this correctly. A Figure would be helpful. But the optics remains unclear. How far was the OLED display away from the objective and where exactly was the trial lens placed? I believe some ray tracing is necessary to determine how much defocus was imposed in the photoreceptor plane in the current optical arrangement. How was it verified at which point the patterns were in best focus? Furthermore, I believe that it is extremely important to control for magnification and luminance changes. It is critical when luminances change with spatial frequencies and defocus (Figure 2). Finally, stationary patterns may not be the best way to uncover the full potential of ganglion cell responses (there is literature on this - note that in a living eye, retinal images move all the time).

In summary, at present, I am not fully convinced that the changes in RGC responses that the author describes are really due to changes in focus. A better description of the optics is necessary.

Specific

line 17. Defocus causes lower contrast and less steep luminance changes. I am not surprised that ON/OFF receptive field  

line 21. That the isolated RGC show more noisy responses is not so surprising because they are no longer synchronized

line 26. “We hypothesize ...” Why is a single author talking as “we”?

line 28. “retinal ganglion cells” not “retina ganaglion cells”

line 30. I have doubts that only ON and OFF alpha RGC /RGCs detect defocus - there may be others also

line 41, delete comma after emmetropization

line 55, add new literature about RGC types in mouse retina (Baden, Euler, Franke etc, after 2017) - there are more

line 65. What are “recognized biological effects”? Describe more concisely what you mean.

line 68. I believe that one should avoid to state that any results are far-reaching ... only future will tell what the big discoveries were

line 71. Better: “differently defocused images were projected ...”

line 76. What is the evidence for an important function of amacrine cells in “increasing the signalling and decreasing the background noise”. Better would also be to write “reducing the signal to noise ratio”

line 78. “Ethics approval”

line 85. Delete “were used in the study” (it appears twice)

line 87. Can you exclude an effect of anesthesia on RGC function?

line 98. For “full” field stimulation, why did not you not use the same OLED display with no pattern?

line 108. Why did you use narrow band light for stimulation and not broadband? There is increasing evidence (see ARVO 2019) that broadband light is necessary for normal function of emmetropization.

line 113. How did you check that the patterns were really focused on the photoreceptors?

Line 116. You wanted to saturate the rod pathway. But note that at least 95% of the mouse photoreceptors are rods. This means that cone sample only at low density and are probably mainly concerned about color vision, but not visual acuity and defocus. Note that mouse mutants without rods have very poor visual acuity, but mutants with only rods have similar visual acuity as wildtype (PubMed PMID: 15623801). So, at the end, you may perhaps be interested in the rods ...

line 137 “intensities varying over 5 log units”

line 148 “where R is ...”

line 157 “tissues were rinsed ...”

line 170. What is a “significant design”?

line 172 better “the magnitude of refractive errors ...”

line 181 - my major problem: if you add a +20D lens to a higher power lens like the objective, its effective will not be 20D - see thick lens formula. It is also very important were the object plane (the OLED) and the image plane (the photoreceptor plane) exactly are. The optics really needs to be fully explained and illustrated.

line 189. There is no evidence of accommodation in the mouse.

I stop here. I would not say that the effort of the author is useless. But I believe that, unfortunately, more effort is necessary to demonstrate an unequivocal effect of defocus on the responses of RGCs.

Author Response

Dear respected reviewer,

I sincerely thank the reviewer for your constructive criticisms and valuable comments. Accordingly, we have addressed all the comments as shown in the revised manuscript with additional interpretations and figures. My responses to the referee’s comments are attached.

Thanks again.

Best

Feng

Reviewer 2 Report

I did enjoy reading the manuscript because the originality and the quality of data presented.  The results clearly indicated that defocussed signal is coded by the retinal ganglion cells and ultimately it may affect the emmetropisation process.  I do have a few questions and they are as follows:

1.       Perceptually it has been shown that a human subject can perceive depth order with chromatic light but not with monochromatic light Vision Research 45(8): 1003-1011.  I wonder if there was any particular reason that the 525 nm wavelength was chosen.

2.       Spatial frequency is affected by the defocussed blur, what is the method to correct for the spatial frequency at different depth planes?

3.       Line 222-225: As the stimulus becomes defocussed, stimulus size changes accordingly and therefore for it affects the regions of stimulation.  By limiting the projected image size to minimise the effect of receptive field I wonder if the surround effect can assist RGC to signal sign of blur.

Author Response

Dear respected reviewer,

We sincerely thank the reviewer for valuable comments. Accordingly, our responses to the reviewer’s comments are attached.

Thanks again.

Best

Feng

Reviewer 3 Report

This is a well-designed study that explores the contribution of retinal ganglion cells to defocus mediated mechanisms of eye growth and refractive error development using a mouse model at a cellular level. It is interesting that some ON and OFF RGCs have responded differentially to positive and negative imposed optical defocus (although some cells have also exhibited a similar response to both types of defocus) and that the retinal amacrine cells have exhibited a potential contribution to this differential response. However, a number of important studies  (particularly from the human eye), that are relevant to this topic, have not been discussed, and the paper would benefit substantially if the results are discussed in relation to these references (included in my comments below). The “methods” and “results” sections would also require significant revision, as they are difficult to understand/follow in the current format. My comments regarding this manuscript are as follows:

Major comments:

What is the functional/physiological differences between the ON or OFF alpha RGCs and the ON-OFF RGCs, particularly in the context of emmetropisation? This needs to be explained briefly in the introduction.

Line 13-15: “A mono green organic light-emitting display controlled by computer, presented different defocused images generated by PsychoPy onto the photoreceptor layer.” This is slightly misleading as it conveys that an image containing simulated blur was imposed on the photoreceptors, while in Methods it is stated that the image plane has been ‘physically’ shifted forward or backward relative to the photoreceptors, thereby creating ‘real’ optical defocus rather than simulated blurred image. The sentence needs to be revised accordingly.

Line 17-20: “ Some OFF alpha RGCs and ON alpha RGCs showed a different response to plus and minus defocused images. ON and OFF units of ON-OFF RGCs also responded differently to defocused images and spatial frequency images.” In what aspect has the responses to plus and negative defocus been different? From the results, it appears that some RGCs have actually exhibited similar responses to both positive and negative imposed defocus.

Line 20-21: “ After application of gap junction blocker, the probability of spikes of RGCs decreased.” Was this observed in the presence of optical defocussed image? Needs clarification in the abstract.

Line 46-47: “The emmetropization process is regulated by defocused image associated with eye growth and refraction (Smith and Hung, 1999, Smith et al., 2013).” There is wealth of information from studies examining the human choroid and retinal bioelectrical activity that have confirmed the findings from animal research that optical defocus (both in short-term – e.g. doi:10.1167/iovs.10-5457, doi:10.1167/iovs.18-24815, doi.org/10.1111/opo.12609, doi.org/10.1016/j.visres.2011.10.017 and long-term – e.g. doi.org/10.1136/bjo.2004.064212, doi:10.1167/iovs.13-11904) influences the mechanisms that would lead to emmetropisation/refractive error development. Suggest to also cite these important evidences from the human eye along with animal studies.

Line 48-49: “It is important to determine whether the retina alone could detect the sign of defocus during this visually-guided refractive development period (Schaeffel and Wildsoet, 2013).” Need to discuss the recent evidence provided by human electrophysiological studies regarding the contribution of the retina in decoding positive and negative defocus (e.g. doi.org/10.1371/journal.pone.0123480, doi.org/10.1016/j.visres.2011.10.017).

Line 68-70: “These novel and far-reaching results support our hypothesis that retinal signal  analysis has great promise for defining the signaling mechanisms modulating refractive development.” The human electrophysiological studies mentioned in the above also provide strong support for this hypothesis.

Line 110-111: “…the area of retina that received light stimuli was 250 μm in diameter.” Should this be patterned light stimuli? Please clarify.

Line 114-115: “The background light intensity was 700 isomerizations Rh*/rod/s and the stimulus was above 3 ×104 Rh*/rod/s.” However, earlier in the text, in Lines 104-105, it is stated that “The intensity of the different light stimuli was above 100 isomerizations (R*)/rod/s in the photopic range.” Also, later in the “results” section, the background light intensity has been reported to be 937 Rh*/rod/s (at line 248).  Please clarify and provide consistent values for the brightness level of the background and stimulus light across the paper.

Line 115-117: What was the reason for saturating the rods and recording the RGC responses from the cone pathway only? Please provide brief justification here.

Line 139-140: Please specify how many RGC cells from each cell category has been studied.  

Line 176-179: “As it has been calculated that axial elongation of 5 μm could induce 1 Diopter (D) refractive error (Schaeffel, 2008), 100 μm defocus would be expected to induce plus or minus 20 diopters refractive error under microscopy depending on the direction of defocus.” The estimation made by Schaeffel using an alert mouse model (with intact anterior optical system) does not appear to be relevant to the work presented here where the anterior optics and the vitreous humour of the mouse eye has been removed.

Lines 199-201: “In addition, ON and OFF α-RGCs could be further visualized by Neurobiotin injection after recording (Figire-1 C and F). Double labeling with anti-200 ChAT antibody showed their large mono-stratified dendritic arbors.” Please provide a brief explanation of the anatomical and physiological/functional differences between the two type of α-RGCs studies here?

Line 207-208: “At these light intensities, the RGCs’ light responses are expected to be activated in the cone pathway where is believed to be responsible for myopia.” This statement needs a reference. Also, how much is the saturation level for the cone pathway (given that the graphs in Figure 1 B and E appear to suggest that the alpha RGCs saturate with light intensities above 100 Rh*/rod/s and the reported light intensities used in the experiment appear to be much higher at 1.53x105Rh*/rod/sec to 1.69 x105Rh*/rod/sec)?

Line 209-225: Quite difficult to understand with multiple grammar errors. Please consider to rewrite these three paragraph and make them easier to understand.

Line 266-269: “By projecting different spatial frequency images, different defocused images in the retina were attempted to be mimicked and RGCs were observed to see whether their light responses were similar between the defocused images and different spatial frequency image.” It should be clarified here that altering the spatial frequency content of the projected image would only simulate image blur without providing any cue to the sign of image blur (given that both myopic and hyperopic defocused images would lose high spatial frequency content regardless of the sign of defocus).

Lines 169-288: The first 4 sub-sections reported in the “Results” appear to be pilot experiments to find out the optimal values for various characteristics of the light stimulus used in the experiment, including the light intensity, stimulus size and spatial frequency content. Suggest to move this information to the “Methods” in a sub-section detailing the characteristics of the light stimulus.    

Line 324-325: “However, the cell only responded to a +5 diopters defocused image (Figure 6-A).” A bit vague. Please reword.

Line 333 and Line 335: Please include the percentage of ON transient RGCs and ON sustained RGCs showing no difference in response to plus and negative defocus.

Line 355: “The peristimulus time histogram of the ON α-RGC showed that the probability of spikes decreased from 0.87 to 0.79 (Figure 7 A-B).” These figures do not match with the Figure 7 caption: “The peristimulus time histogram of the ON α-RGC showed that the probability of spikes decreased from 0.96 to 0.79.” Please correct. Please ensure the numbers presented in text match up with those presented in graphs and Figure captions.

Line 385-393: It seems to me that section 7 and 8 in the “results” are discussing the same thing, that is application of gap junction blocker affects the ON and OFF RGCs. Please clarify (and merge these two sections if they are discussing the same aspect of RGC responses to defocus).

Line 473-474: “Although several interventions, including orthokeratology, atropine therapy and mixed defocused lenses can slow down axial growth significantly …” please provide reference for this.

Line 479-480: “The evidence from animals implicated the role of the visual image on the retina in 479 refractive development as the main controlling mechanism (Angi et al., 1993, Norton, 1999).” It seems that Angi et al 1993 is not a proper reference for this sentence. This paper does not provide evidence from animal models of myopia!

Line 485-486: Some important references from human studies showing bi-directional eye length and choroidal thickness changes with optical defocus are missing here including:   doi:10.1167/iovs.10-5457,  doi:10.1167/iovs.18-24815, doi: 10.1097/OPX.0000000000000035,  doi.org/10.1016/j.exer.2012.08.002

Line 486-487: there is also a recent evidence from the human eye suggesting a localised choroidal response and retinal response to defocus that both need to be discussed here: doi.org/10.1111/opo.12609, doi.org/10.1371/journal.pone.0123480

Line 487-498: The fact that some RGCs have exhibited differential response to positive and negative imposed defocus should be further elaborated here. Given that other animal studies suggest that ON RGCs may contribute to anti-myopiagenic mechanisms and OFF RGCs may contribute to myopiagenic mechanisms, the author needs to further discuss the findings of this study in relation to these previous animal studies. There is also a recent interesting evidence from the human eye regarding the contribution of ON/OFF retinal pathways to the mechanisms of emmetropisation that should be included in the discussion here (doi.org/10.1038/s41598-018-28904-x).

Line 514-516: It would be great to elaborate more on this in relation to the recent evidence suggesting human retinal response variations with optical defocus and spatial frequency content of the stimulus (doi.org/10.1111/opo.12609, doi.org/10.1371/journal.pone.0123480).

Minor comments:

Line 98-99: “A green (525 nm) light-emitting diode delivered uniform full-field visual stimulation on the 98 surface of the retina.” This is repeated in Lines 105-106. Please correct.

Line 26: A typo here: ‘hypothesis’ should be ‘hypothesise’

Line 84-85: “were used in the study” is repeated twice.

Line 323: “The probability of spikes of OFF RGC responses …” should this be transientOFF RGC?

7 of 29 (24%) OFF transient RGCs showed similar responses

Figure 1 caption: “ON and OFF RGCs show increasing spikes to a full-field 525 nm light stimulus”. Should this be “ON and OFF RGCs show increasing spikes with increase in the light intensity of a full-field 525 nm light stimulus”?

Figure 2 A: Please use a more typical approach to report large numbers (rather than using e+) on the y axis (e.g. replace 3.0E+04 with 3 × 104), consistent with the numbers reported in the Figure 2 caption.

Figure 2 B: Please clarify what “diameter” refers to? I suspect it is the diameter of the projected stimulus light on the photoreceptor layer? Also there is a typo in the legend “diameter”

In the caption of Figure 2, please clarify that the light intensity/spot size are reported for an image projected on the photoreceptor layer with different magnitudes of optical defocus. Also, the caption needs to be revised to be easier to understand.

Figure 3: Suggest to use a consistent scale for the y-axis across the three time histograms illustrated in Figure 3 A-C. Also, please clarify what does the horizontal dashed line in each time histogram indicate?

Figure 4: Suggest to use a consistent scale for the y-axis across the time histograms used in this Figure (and also the time histograms presented in Figures 5, 7, and 9).

Figure 4 caption: “D: OFF α-RGCs lost their light response to 125 μm, 287 0.0067 cycles/degree light stimulation.” Shouldn’t this be the response from an ON α-RGC rather than an OFF α-RGC?

Figure 6: In the legends presented in each graph, “OFF Ganglion Cell” should be replaced by “transient OFF ganglion cell”, consistent with the figure caption.

Figure 6 caption: “The cells showed no response to defocused image more than +5D (red triangle) and less than -20D (blue triangle).” Based on the graph A, red triangle represents the cut-off for cells not responding to less than -20D. Also, in line 345-346, “The cells showed no response to defocused images more than +16D (blue triangle) and less than -20D (red triangle)”, there are no red and blue triangles in graph B. Please correct the caption to be consistent with labels shown in the graphs.  

Author Response

Dear respected reviewer,

I sincerely thank the reviewer for your constructive criticisms and valuable comments. My responses to the referee’s comments are attached.

Thanks again.

Best

Feng

Round 2

Reviewer 1 Report

The author has responded to all my comments in detail.

I still would like to express some reservation about the conclusions derived in paragraph 6, and in Figure 6. Figure 6 seems to suggest that the peak responses of the different RGCs occur at different levels of defocus. However, the x-axis is different in (A) and (B)-(D), suggestion a different peak location. Given the width of the red curves, it is not fully convincing that the cells really respond differently to positive and negative defocus. Note that the peaks are at about +2D in (A), +2D in (B), -2D in (C) and about 0D in (D). Any chance of statistics that they are really different? Given the depth of focus of the mouse eye of 10D or more, are these max 4D differences really significant? That the cell responses level off at high amounts of defocus is kind of trivial since only low spatial frequencies are presented which don't stimulate receptive field with antagonistic ON/OFF structures.

In summary, I have no objections against showing these data but perhaps the author can reconsider whether his statement "RGCs respond differently to plus and minus defocus" is really supported by the data.

Author Response

(The authors gave the same response as above.)
